# Characteristics of Sensory Neuron Dysfunction in Amyotrophic Lateral Sclerosis (ALS): Potential for ALS Therapy

**DOI:** 10.3390/biomedicines11112967

**Published:** 2023-11-03

**Authors:** Soju Seki, Yoshihiro Kitaoka, Sou Kawata, Akira Nishiura, Toshihiro Uchihashi, Shin-ichiro Hiraoka, Yusuke Yokota, Emiko Tanaka Isomura, Mikihiko Kogo, Susumu Tanaka

**Affiliations:** Department of Oral and Maxillofacial Surgery, Graduate School of Dentistry, Osaka University, 1-8 Yamadaoka, Suita 565-0871, Osaka, Japan

**Keywords:** amyotrophic lateral sclerosis (ALS), sensory neuron, ALS mouse model, neurodegeneration, dorsal root ganglion (DRG), mesencephalic trigeminal nucleus (MesV)

## Abstract

Amyotrophic lateral sclerosis (ALS) is a devastating neurodegenerative disorder characterised by the progressive degeneration of motor neurons, resulting in muscle weakness, paralysis, and, ultimately, death. Presently, no effective treatment for ALS has been established. Although motor neuron dysfunction is a hallmark of ALS, emerging evidence suggests that sensory neurons are also involved in the disease. In clinical research, 30% of patients with ALS had sensory symptoms and abnormal sensory nerve conduction studies in the lower extremities. Peroneal nerve biopsies show histological abnormalities in 90% of the patients. Preclinical research has reported several genetic abnormalities in the sensory neurons of animal models of ALS, as well as in motor neurons. Furthermore, the aggregation of misfolded proteins like TAR DNA-binding protein 43 has been reported in sensory neurons. This review aims to provide a comprehensive description of ALS-related sensory neuron dysfunction, focusing on its clinical changes and underlying mechanisms. Sensory neuron abnormalities in ALS are not limited to somatosensory issues; proprioceptive sensory neurons, such as MesV and DRG neurons, have been reported to form networks with motor neurons and may be involved in motor control. Despite receiving limited attention, sensory neuron abnormalities in ALS hold potential for new therapies targeting proprioceptive sensory neurons.

## 1. Introduction

Amyotrophic lateral sclerosis (ALS) is a neurodegenerative disease associated with protein abnormalities and neuritis [1,2,3,4]. Genes encoding superoxide dismutase 1 (SOD1), chromosome 9 open reading frame 72 (C*9*orf*72*), fused in sarcoma (FUS), TANK-binding kinase 1 (TBK1), and TAR DNA-binding protein (TARDBP) are known causative factors in familial ALS, accounting for 5–10% of patients with ALS [1,5,6,7,8,9,10]. Most ALS cases are sporadic ALS, and most patients with ALS show intracytoplasmic aggregation of ALS-inducing factors such as SOD1 and TAR DNA-binding protein 43 (TDP-43) in motor neurons, which is increasingly considered a feature of ALS [2,11,12,13,14,15]. The clinical phenotype of ALS includes both upper motor neurons (UMNs) and lower motor neurons (LMNs), but non-motor areas are increasingly found to be involved, leading to non-motor symptoms that include autonomic dysregulation, cognitive impairment, sensory impairment, and frontotemporal dementia (FTD) [15,16,17,18,19,20,21,22]. Clinical and preclinical studies have reported abnormalities in sensory neurons, and further research in this regard is ongoing. Several potential protein biomarkers have also been reported [23,24,25,26,27,28,29,30,31], but sensory neuronal abnormalities in ALS remain to be elucidated in detail. Proprioceptive sensory neurons not only transmit information from muscle spindles to the brain but also act on motoneurons and are involved in the control of muscle movement. Abnormalities in proprioceptive sensory neurons, such as the dorsal root ganglion (DRG) [8,32,33,34] and mesencephalic trigeminal nucleus (MesV) [35,36], were reported early in ALS research and may represent novel targets for ALS treatment.

This review summarises the latest research on sensory neuron abnormalities, with a focus on clinical and preclinical studies. In addition to previous reports [2,16,19], we discuss sensory neuron abnormalities and propose new therapies for ALS, targeting proprioceptive sensory neurons.

## 2. An Overview of ALS

### 2.1. Clinical Characteristics of ALS

ALS is a progressive neurodegenerative disorder characterised by motor neuron degeneration in the brain and spinal cord [32,37,38]. It primarily affects voluntary muscles, leading to muscle weakness, atrophy, and eventually, paralysis [27,39,40,41,42]. Although the exact cause of ALS is not fully understood, it is believed to be a combination of genetic and environmental factors [43,44]. In some cases, specific genetic mutations that are involved in survival have been identified, including the homeostatic iron regulator (HFE) p.H63D polymorphism, a modifier of the SOD1 gene [45]. However, most cases are sporadic, with no known genetic cause [46]. While ALS is currently incurable, various treatment options are being studied to manage its symptoms and improve the quality of life of individuals with the disease [47,48,49]. These treatments include medications to manage muscle cramps and spasticity, assistive devices to aid mobility and communication, and multi-disciplinary care involving physical, occupational, and speech therapy. In recent years, the disease paradigm has changed and ALS is increasingly being considered a multi-system disease rather than a motor-neuron-limited disease.

### 2.2. Motor Neuron Dysfunction in ALS

ALS is characterised by a progressive loss of both UMNs and LMNs [50,51]. UMNs are located in the motor cortex of the brain, whereas LMNs are found in the brainstem and spinal cord. The dysfunction and degeneration of these neurons leads to the characteristic muscle weakness, atrophy, and spasticity seen in ALS [51,52]. The exact mechanisms underlying motor neuron dysfunction in ALS are unclear. However, several key pathological features have been identified. One important protein involved is TDP-43, which forms abnormal aggregates in the cytoplasm of affected neurons and disrupts normal cellular processes [13,53]. Motor neuron dysfunction in ALS is associated with disturbances in the transmission of signals between neurons and muscles. These disturbances result in the failure of motor neurons to adequately stimulate muscle contractions, leading to muscle weakness and the eventual loss of voluntary movement control. The dysfunction and degeneration of motor neurons in ALS are thought to be influenced by various factors, including oxidative stress, mitochondrial dysfunction, impaired protein degradation pathways, glutamate excitotoxicity, and inflammation [54,55]. These factors contribute to the progressive loss of motor neurons and clinical manifestations of ALS.

## 3. Clinical Research in Patients with ALS

### 3.1. Clinical Features of Sensory Neuron Dysfunction in ALS

In recent years, growing evidence indicates that sensory neuron dysfunction contributes to the clinical manifestations of ALS and influences disease progression and prognosis [17,56,57,58,59,60,61] (Table 1). Liu et al. reported that sensory impairments are often mild and there is no significant progression [62]; however, Hammad et al. reported that a third of patients with ALS experience sensory symptoms, with corresponding nerve changes, particularly affecting large, myelinated fibres [56]. Sensory manifestations include paraesthesia, sensory loss, neuropathic pain, and heightened sensitivity to external stimuli [62,63]. Patients may experience these symptoms in various anatomical distributions, including the upper and lower extremities, trunk, and face. The coexistence of sensory symptoms with motor symptoms can significantly impact the overall disease burden and quality of life in patients with ALS [56]. In addition to self-reported symptoms, objective sensory examination findings support the presence of sensory neuron dysfunction in ALS. Sensory deficits, such as impaired vibration sense, reduced proprioception, and decreased thermal and pain sensation, have been observed in patients with ALS [64,65,66]. Neurophysiological studies have also demonstrated abnormalities in sensory nerve conduction parameters, including prolonged sensory nerve action potentials and reduced sensory nerve action potential amplitudes [67,68,69,70,71,72].

### 3.2. Peripheral Nerve Abnormalities Observed in Patients with ALS

In recent years, several studies have shed light on the involvement of sensory nerve conduction abnormalities in ALS (Table 1). A study by Heads et al. uncovered noteworthy findings, revealing axonal atrophy, increased remyelination, and abnormalities in small-diameter nerve fibres in patients with ALS [78]. These changes correlate with the duration of the disease [78]. Subsequent investigations further emphasised the significance of sensory nerve conduction abnormalities in ALS. Gregory et al. reported that the potential amplitude of sensory nerve conduction was significantly reduced in 19 patients with ALS compared to healthy control individuals, despite normal nerve conduction velocities [67]. Nolano et al. added more depth to the understanding of ALS-related sensory issues when they examined 41 patients with sporadic ALS and observed a loss of epidermal nerve fibres and Meissner bodies. Interestingly, these findings were associated with a partial reduction in cutaneous vascularity and were correlated with disease progression [60]. Distinguishing between different ALS subtypes, Truini et al. noted that patients with spinal cord-onset ALS exhibited distal small-fibre neuropathy characterised by abnormal heat pain thresholds and decreased epidermal nerve fibre density, in contrast to patients with early-onset ALS [75]. Hammad et al. conducted a study involving 103 patients with ALS that revealed that nearly 30% of these patients exhibited sensory symptoms and abnormalities in leg sensory neuron conduction studies [56]. Peroneal nerve biopsies from these patients showed histologic abnormalities, including the loss of large-calibre myelinated fibres, axonal loss, and regeneration [77].

Overall, these studies collectively suggest that sensory nerve conduction abnormalities are a notable feature of ALS. Furthermore, recent research on Asian patients with ALS and the identification of abnormal TDP-43 aggregations in skin-derived fibroblasts from patients with sporadic ALS by Riancho et al. support the notion that ALS encompasses a range of neurodegenerative disorders, including sensory neuropathy [79]. Additionally, there is growing interest in the potential utility of distal sensory nerve conduction studies as a diagnostic tool in ALS research.

### 3.3. Pathological Alterations in Sensory Neurons in Patients with ALS

Sensory peripheral nerve conduction has been a subject of extensive study in the context of ALS. Liu et al. conducted research on 150 patients, employing pathological tests, and detected affected sensory fibres in 22 of these patients, accounting for approximately 14.7% of the cohort [62]. In a retrospective investigation involving 17 patients with ALS who had undergone a peroneal nerve biopsy, a significant finding emerged: more than two-thirds of these patients exhibited substantial axonal loss [80]. Moreover, in a separate study involving 22 patients, Hammad et al. reported histologic abnormalities in 91% of cases. These abnormalities included the loss of myelinated fibres, predominantly those of a large calibre, accompanied by axonal loss and regeneration [56]. In a study by Iglesias et al. on 21 patients with moderate ALS without clinical sensory signs (mean ALS Functional Rating Scare (ALSFRS) score = 39.3 ± 1.0), spinal diffusion tensor imaging reported anatomic damage to ascending sensory fibres in approximately 60% of the patients. They concluded that sensory neuropathy is underestimated in patients with early ALS [76]. Another prospective study, encompassing 32 patients with ALS, analysed both nerve conduction and peroneal nerve biopsies; while nerve conduction did not reveal specific sensory abnormalities, the histological analysis uncovered abnormal axonal swelling in all patients with ALS [58]. Importantly, TDP-43 staining was negative, suggesting failed regeneration in small sensory nerve fibres [58]. However, Ren et al. performed skin biopsies on 18 patients with ALS and reported that TDP-43 deposits were present in cutaneous sensory and autonomic nerves in 33.3% of these patients [74].

### 3.4. Abnormalities of the Somatosensory Cortex in Patients with ALS

The concept of somatosensory cortex overexcitation in ALS is intriguing because of its potential impact on the survival of patients with ALS. This hypothesis proposes that somatosensory cortex overexcitation during certain stages of the disease may be a compensatory mechanism used by the sensory cortex in response to reduced motor function [81]. In ALS research, somatosensory evoked potentials (SEPs) have been used to assess sensory signal transduction along central spinal cord pathways. SEPs reveal changes in patients with ALS, with a reported incidence of 50% [73]. Notably, these SEP changes tend to progress relatively slowly, suggesting that, compared to the rapid progression of motor symptoms, sensory deficits may progress at a more gradual pace, despite their common presence at diagnosis [82]. Shimizu et al. reported that larger somatosensory cortex amplitude SEPs correlated with significantly shorter survival times in 145 patients with sporadic ALS. This observation raises the possibility that somatosensory cortex overexcitation may be associated with ALS progression and impacts patient survival [71]. Late SEPs, which reflect cognitive-motor pathways, also show significant reductions in patients with ALS. This observation underscores the multifaceted nature of sensory pathway involvement in ALS [19].

A recent study by Nardone et al. has enhanced our understanding of this phenomenon [73]. The aim of this study was to assess sensory cortical function in 20 consecutive patients with sporadic ALS by evaluating high-frequency SEPs [73]. The results revealed that somatosensory cortex de-repression in patients with ALS becomes more pronounced after the second year of disease progression, suggesting that changes in somatosensory cortical activity may evolve over and contribute to the course of the disease [73].

More recently, laser evoked potentials (LEPs) have emerged as a new tool to assess central pain signal conduction; studies using LEPs have revealed abnormal latency delays in patients with ALS [83]. These findings suggest the possibility of degeneration in subcortical sensory structures, further highlighting the complex nature of altered sensory pathways in ALS.

In summary, somatosensory cortex hyperexcitability appears to predict shorter survival in patients with ALS. In conclusion, sensory cortical hyperexcitability may reflect a multisystem neurodegenerative disease or represent a compensatory upward control mechanism associated with motor cortical dysfunction.

## 4. Preclinical Research in ALS

### 4.1. Pathological Alterations in Sensory Neurons in Patients with ALS

Recent studies have demonstrated that sensory neurons, including those involved in proprioception and nociception, are also impacted in ALS [2,35,72,84]. This broad neuronal involvement may contribute to the diverse clinical manifestations observed in ALS. Pathological alterations in sensory neurons in ALS encompass the degeneration of DRG neurons [34,84], axonal degeneration, demyelination, protein aggregation, mitochondrial dysfunction, gene abnormalities, and glial activation [85,86] (Figure 1). ALS preclinical studies have been conducted using SOD1G*93*A and TDP-43 A*315*T transgenic mice and dSOD1G*85*R and C*9*orf*72* transgenic *Drosophila* (Table 2); for example, in TDP-43A*315*T and SOD1G*93*A ALS mice, DRG neurons have slower neurite outgrowth and fewer branches than those in control mice. They are also more sensitive to vincristine treatment, which causes axonal degeneration [84]. In another example, Ringer et al. used SOD1G*93*A ALS mice and elucidated that the dendritic projection pathology of glutamatergic sensory neurons, restricted to olfactory bulb mitral cells and retinal nodal cells, is responsible for abnormalities in olfaction and vision [87].

### 4.2. Degeneration of DRG Neurons

The DRG is a cluster of sensory neurons located along the spinal cord. As the reflex arch circuit is created by dorsal root nerves joining the motor nerves via interneurons within the spinal cord, the innervation of dorsal root nerves is crucial to its construction. Rubio et al. reported the neuronal cell bodies of the DRG and axon terminals of the skin and determined that the number of DRG neurons remained unchanged, although the density of nerve fibres, especially nonpeptidergic fibres, was decreased in the epidermis [78] (Table 3). This finding indicates axonal involvement, particularly distal axonal involvement, as the neuronal cell bodies remained intact [78]. Guo et al. reported that DRG axons are severely damaged in SOD1G*93*A mice, and although no obvious reduction in the number of DRG neurons is observed, many DRG neurons in SOD1G*93*A mice have numerous vacuoles. Furthermore, images acquired through transmission electron microscopy reveal mitochondrial swelling in numerous DRG neurons in SOD1G*93*A mice [90]. The DRG in mice injected with serum from patients with ALS demonstrate decreased Ca currents through high-voltage-activated Ca channels [91]. In a study using ALS model mice (TDP-43A*315*T and SOD1G*93*A), TDP-43 and SOD1 were found to directly affect and sensitise DRG neurons to stress [84]. Levels of the stress-related molecules activating transcription factor 3 (ATF3) and protein kinase RNA-like ER kinase (PERK) differed between the two mutants [84]. These findings suggest the direct impact of ALS-related factors on sensory neurons, implying their role in ALS pathogenesis [84]. Furthermore, the DRG in SOD1G*93*A mice showed axonal stress features and the accumulation of a peripherin splice variant named peripherin 56 [34]. These results suggest a molecular mechanism for the small-fibre pathology found patients with ALS, wherein the dysfunction of DRG neurons mirrors some of the dysfunctions found in motor neurons [33,92]. In addition to neuronal loss, sensory neurons in ALS exhibit axonal degeneration and demyelination [77], owing to the loss of the protective myelin sheath surrounding the axons, resulting in the impaired conduction of sensory neurons. Such axonal abnormalities are characterised by the presence of axonal swellings and varicosities [77].

### 4.3. Characteristic Abnormalities in Mesencephalic Trigeminal Neurons

MesV neurons are proprioceptive sensory neurons that control mastication [35,96]; these directly affect motor neurons through α-amino-3-hydroxy-5-methyl-4- isoxazolepropionic acid and N-methyl-D-aspartate (NMDA) receptors, which are excitatory postsynaptic receptors for neurotransmitters [97,98]. A study using neonatal ALS model mice (SOD1G*93*A) reported firing inhibition as an electrophysiological feature of the MesV in neonates [35] (Table 3). This inhibition is caused by a decrease in the Na current due to a deficit in the Nav1.6 Na channel, in contrast to the increased Na current observed in motor neurons [99]. Furthermore, the addition of Na channel-derived model currents in the dynamic clamp method improves the firing anomaly characteristics of the MesV. These findings suggest that the abnormalities in the MesV of neonatal ALS model mice may be due to the dysfunction of Na channels [35]. Abnormal firing of the MesV is also observed in adult SOD1G*93*A mice [36]. Furthermore, abnormalities in masticatory rhythms correlate with weight loss in adult SOD1G*93*A mice [36]. These findings suggest a novel reflex circuit-specific sensory issue, potentially linked to muscle twitching, which could aid in early diagnosis and treatment strategies for ALS [61,100].

### 4.4. Protein Aggregates and Mitochondrial Dysfunction

Similar to motor neurons, sensory neurons in ALS display abnormal protein aggregation. Aggregates of misfolded proteins, such as TDP-43, accumulate within the cell body and axons of sensory neurons [94]. Activating bone morphogenetic protein (BMP) signalling in non-motor and motor neurons improves motor function in the early and late disease stages. Non-motor neurons contribute to early ALS-induced motor dysfunction, suggesting potential therapeutic targets for delaying motor neuron degeneration [72]. Furthermore, mitochondrial dysfunction is a common feature in ALS, which leads to impaired energy production and neuronal viability [101,102]. For example, a 2009 study using the SOD1G*93*A mouse model showed that the axoneme of mild DRG neurons was occupied by vacuoles and swollen mitochondria, with mild degeneration. Histological features of degeneration were seen as early as 60 days of age, after which the degeneration gradually worsened. Thus, the histological abnormalities occurred long before the animals showed signs of disease. Mitochondrial damage and the caspase-mediated apoptosis of spinal motoneurons have also been shown to play an important role in motoneuron degeneration in hSOD1G*93*A mice [61,103]. In a study using the *Drosophila* knock-in model of ALS, the reduced number of mobile mitochondria in the axons of multidendritic neurons were found to be most likely due to an inborn defect in the cell organelles themselves, rather than a defect in the axonal transport mechanism [104]. Diseased sensory neurons display specific mitochondrial shape issues and increased synapse mitophagy, which are reversed upon the down-regulation of dynamin-related protein 1 (Drp1). Adjusting oxidative phosphorylation (OXPHOS) subunits also reverse ALS-related mitochondrial defects [104]. Further analysis revealed disruption in the electron transport chain, indicated by genetically encoded redox biosensors. Sensory neurons exhibited compartment-specific abnormalities in mitochondrial morphology, with increased mitophagy at the synaptic regions [90]. Finally, our study on a SOD1-knock-in ALS model revealed compartment-specific mitochondrial abnormalities in sensory neurons, shedding light on their role in ALS and emphasising that mitochondrial transport issues may originate from intrinsic dysfunction rather than trafficking machinery problems [104]. Our work also demonstrated that correcting OXPHOS subunit genes and balancing mitochondrial fission/fusion can restore abnormal morphology to normal levels [104]. These findings pinpoint potential therapeutic targets, advancing our understanding and aiding in ALS treatment (Table 3).

### 4.5. Glial Involvement

Glial cells, including astrocytes and microglia, play a significant role in ALS pathogenesis. Pereira et al. established a cultured human sensorimotor organoid model and suggested that glial cells may be involved in motor neuron atrophy in ALS [105]. However, the extent to which astrocytes and microglia exert non-cell autonomous effects or even contribute to the capacity of vascular cells for extensive angiogenesis and the blood–brain barrier is unclear [105]. Though glial cells store lipid droplets as an energy reserve, emerging evidence links lipid droplets to neuron–glia metabolic interactions [106]. The high lipid peroxidation levels observed in ALS may serve as a protective detoxification mechanism against oxidative stress in sensory neurons, which is relevant to ALS, given the prevalent lipid metabolism abnormalities in patients with ALS and SOD1 mice. Recent findings also suggest a metabolic shift from glucose to lipids as an energy source in SOD1 mouse ALS models. If this occurs in the DRG of SOD1 mice [107], SGCs might store excess neutral lipids as a neuroprotective strategy to reduce lipotoxicity. Intriguingly, Ruiz-Soto et al. detected lipid droplets within autolysosomes, suggesting a mechanism of lipophagy [8]. Consistent with this finding, Rudnick et al. observed autophagy activation in motor neurons during ALS progression in SOD1G*93*A mice, and we frequently observed autophagosomes in SGCs from this ALS mouse model [108].

In ALS sensory neurons, satellite glial cells (SGCs) have been examined; SGCs at the DRG showed abnormal SOD1 accumulation, reinforcing the pathogenic role of glial cells in ALS [8]. These changes lead to lysosomal storage issues and occasional degeneration. This finding highlights SGCs as key targets in ALS pathology, underscoring the role of glial cells in motor neuron diseases [8]. In SGCs, impaired lysosomal accumulation due to oxidative stress is observed, suggesting a novel potential mechanism contributing to ALS pathogenesis. SGCs also contribute to the normal function of sensory neurons, highlighting the importance of astroglial cells [8]. Based on the sensory–motor network and prion hypothesis, it is suggested that SGCs may function as a cause of sensory symptoms and influence motor neuron pathogenesis. In ALS spinal cords, reactive astrocytes over-express γ1 laminin, particularly around the affected regions [109]. This unique γ1 laminin distribution indicates that astrocytes may produce γ1 laminin to aid the survival of neurons in ALS, whereas TDP-43 is lost in the astroglia [109,110]. The multiple ways in which γ1 laminin and its KDI domain protect neurons neurotrophically and enhance their viability suggest that the overexpression of γ1 laminin by reactive astrocytes in the ALS spinal cord may be an attempt by the glia to protect neurons and combat ALS pathology. In a study conducted by Peng et al., these changes do not affect motor neurons or the neuromuscular junction, but the number of mature oligodendrocytes is selectively reduced, implying that the loss of TDP-43 affects oligodendrocyte function [110]. An astroglial TDP-43 deletion in mice causes motor deficits, highlighting its importance in maintaining astrocytic protection for motor function [110]. Glial dysfunction may thus contribute to the degeneration of sensory neurons and the propagation of ALS pathology.

### 4.6. Excitotoxicity and Calcium Dysregulation

Excitotoxicity and calcium dysregulation are emerging as key contributors to sensory neuron dysfunction, and their hyper-activation can impair LMNs in ALS [100]. In *Drosophila* sensory neurons, the cytosolic calcium-calpain-A-importin pathway was identified as a regulatory mechanism underlying the nucleo-cytoplasmic transport of TDP-43. The calcium-calpain-A-importin α3 pathway is a therapeutic target of ALS [88,111]. Stracher reported that calpain proteolysis, either directly or indirectly, is associated with neurological diseases, such as ischemia, nerve damage, and spinal cord injury [111]. Calpain may be involved, although not directly, in neurodegenerative diseases, including Alzheimer’s disease, Parkinson’s disease, multiple sclerosis, Huntington’s disease, and ALS [111]. Dysregulated calcium handling and mitochondrial dysfunction have also been shown to collectively disrupt the function of sensory neurons and contribute to their degeneration [88,93,100]. The interaction between the endoplasmic reticulum (ER) and mitochondria significantly impacts neuronal functions; the malfunctioning of the ER–mitochondria interface, the mitochondria-associated ER membrane (MAM), leads to stress, calcium imbalance, and energy disruption. MAM disturbances might initiate axonal degeneration [93], and the MAM is therefore a potential therapeutic target for ALS.

### 4.7. Oxidative Stress and Protein Homeostasis

Oxidative stress and the generation of reactive oxygen species play an important role in neuronal dysfunction in ALS [14,101,112]. In their 2021 study, Zuo et al. reported that TDP-43 aggregation in neurons of mouse and human origin causes susceptibility to oxidative stress [14]. Aggregated TDP-43 sequesters specific microRNAs (miRNAs) and proteins, increasing the levels of some proteins and functionally depleting others. Many of these functionally disrupted gene products are mitochondrial proteins encoded in the nuclear genome, whose dysregulation causes an overall mitochondrial imbalance and increased oxidative stress. This stress–aggregation cycle may underlie the onset and progression of ALS [14]. Dilliott et al. reported that the J protein DnaJC7 may be regulated in response to cellular stress via the heat shock response [113]. DnaJC7 may be essential under other certain stress conditions, such as oxidative stress, protein misfolding stress, and impaired protein quality control in aging cells, especially in motor neurons [113]. Therefore, an understanding of the mechanisms underlying these processes may provide valuable insights into the maintenance of sensory function in patients with ALS and may help in developing therapeutic strategies for improved clinical outcomes.

Protein homeostasis, or proteostasis, networks are important mechanisms that maintain proper protein folding, degradation, and repair within the cell. A growing body of evidence suggests that the TDP-43-induced reduction of stathmin 2 (STMN2) protein and impaired proteostasis play an important role in ALS pathogenesis [113,114]. In their 2022 study, Krus et al. used STMN2 knockout mice to investigate the possibility that STMN2 deficiency is a cause of ALS; the loss of STMN2 results in the loss of motor neurons and axons, leading to motor behaviour deficits and early sensory neuropathy, which are particularly problematic in ALS [114]. Removing STMN2 in motor neurons causes similar problems. Heterozygous mice mimic the pathophysiology of patients with ALS and exhibit motor-selective neuropathy. Thus, the reduction of STMN2 due to TDP-43 abnormalities likely influences the pathogenesis of ALS [114]. Using a *Drosophila* model with ALS-related mutation in all cells, Held et al. observed motor deficits in both the larval and adult stages [72]. Motor neuron changes could not explain the early-stage locomotion decline [72]. A sensory feedback issue in proprioceptor neurons, critical for muscle status relay, was identified [72]. Activating BMP signalling in these neurons rescue motor defects, which is indicative of their role in ALS progression [72]. This finding emphasises the significance of non-motor neurons, potential early ALS biomarkers, and novel therapeutic targets [72]. An understanding of how neural circuit dysfunctions contribute to neurodegeneration is crucial for earlier ALS diagnosis and intervention.

### 4.8. Protein Changes in Sensory Neurons

Studies involving sensory pathways in SOD1 transgenic mice, the most-used mouse model of ALS, have demonstrated pathological changes in the cell bodies of axons of the posterior column, spinal dorsal horn, dorsal root, and DRG neurons [90,115], similar to those observed in Wallerian degeneration. In addition, the abnormal accumulation of SOD1 protein associated with abnormalities in the SOD1 gene in the cell bodies of sensory axons and DRG neurons has been reported [116,117]. Notably, one ALS study also showed increased thickness in certain regions of the retina [118], elucidating a critical role for RAN binding protein 2 (Ranbp2) in signalling between microglia and retinal ganglion cells in the immune response in ALS [119]. Ranbp2 is a protein involved in nucleocytoplasmic transport, and its regulation is impaired in both sporadic and familial ALS [120]. The widespread impairment of the olfactory pathway suggests a common mechanism with classical motor neuron degeneration that goes beyond the pathophysiology of TDP-43. For example, senataxin, a protein associated with dominant juvenile ALS (ALS4) through gain-of-function mechanisms, is most abundantly expressed in the hippocampus and olfactory bulb [16]. TDP-43 has recently been shown to regulate the axonal transport of ribosomal protein mRNA and local translation by ribosomes, which is essential for maintaining axonal morphological integrity [121], and the accumulation of TDP-43 associated with TARDBP gene abnormalities has been reported in case reports [122]. Liao et al. identified eight mutations in the kinesin family member 1A (KIF1A) gene in 10 of 941 patients with ALS; they reported that patients with ALS carrying rare damage variants in KIF1A tend to exhibit sensory deficits and that KIF1A causes hereditary sensory and autonomic neuropathies, which affect sensory neurons [123].

### 4.9. Role of Cytoskeletal Dysregulation and Neuronal Transport

The dysregulation of the cytoskeleton and neuronal transport is also implicated in ALS pathogenesis [124,125]. Schäfer et al. used progressive motor neuropathy model mice carrying a missense loss-of-function mutation in tubulin-binding cofactor E (TBCE) and showed that sensory neuropathy in ALS is a cytoskeletal defect [85]. Liao et al. investigated the genetic association of the KIF1A gene, which encodes a motor protein involved in the transport of vesicles in neurons, based on the whole-exome sequencing of samples from patients with ALS, and identified rare damage variants in KIF1A associated with the disease [123]. The patients with ALS carrying variants tended to exhibit sensory disturbances [123]. A functional analysis revealed that these variants enhanced the binding of specific vesicles to the KIF1A protein [123]. Additionally, the expression of disease-related KIF1A mutants in cultured neurons resulted in an increased co-localisation of vesicles with the KIF1A motor [123]. These findings highlight the involvement of KIF1A-mediated transport in ALS pathogenesis and suggest its role as an important player in the genetic complexity of ALS. Patients with ALS and model mice exhibit neurofilament accumulation in the cell bodies and axons of motor neurons. Abnormalities in neurofilament transport have also been observed in multiple lines of mutant SOD1 transgenic mice, along with reduced tubulin transport in some cases [126]. Overall, these findings suggest that axonal transport abnormalities contribute to motor neuron pathology in ALS mouse models.

## 5. Targeting Proprioceptive Sensory Neurons as a Potential Therapy for ALS

### 5.1. Current ALS Drugs

There are currently no fundamental ALS drugs. The most used is riluzole, which inhibits glutamate release, blocks postsynaptic NMDA and kainic acid-type glutamate receptors, and inhibits membrane potential-dependent sodium channels [127]. The neuroprotective drug riluzole has a modest survival benefit, but has long been the only disease-modifying therapy for ALS [128]. Edaravone (a free radical scavenger) treatment significantly delays disease progression in ALS, with an increasing number of countries approving its use since its initial approval in 2015 [129]. Edaravone has been used to treat acute strokes for almost two decades in Japan; it was approved for the treatment of ALS in Japan and South Korea in 2015; the US Food and Drug Administration (FDA) approved the drug in 2017 and the Chinese National Medical Products Administration approved it in 2019; and it has been used for the treatment of ALS for almost 20 years in Japan and South Korea. In a phase II study, 20 patients with ALS received either 30 or 60 mg of edaravone and a significant reduction in the revised ALSFRS (ALSFRS-R) score was observed during the 6-month treatment period [130]. In a randomised, double-blind phase III trial, significantly smaller reductions in ALSFRS-R scores were also observed compared to the placebo group [131]. However, the exact mechanism of action of edaravone in ALS remains unclear. It demonstrates both neuroprotective effects against oxidative stress and anti-inflammatory effects against activated microglial cells. However, these effects only marginally prolong the life of patients with ALS. Other potential ALS drugs, such as sodium phenylbutyrate and taurursodiol (PB-TURSO), were conditionally approved in Canada in 2022; patients who were randomised to receive PB-TURSO (*n* = 87) had a 47% lower risk of major events compared to the placebo group (*n* = 48), suggesting that early PB-TURSO treatment may significantly slow disease progression and prolong survival in ALS [132]. In April 2023, the FDA approved Tofersen for the treatment of adult patients with ALS associated with a confirmed SOD1 gene mutation [127]. Tofersen is an antisense oligonucleotide that suppresses the production of the harmful SOD1 protein ALS variant by binding to mRNA associated with the SOD1 gene [128]. Finally, the Japan Early-Stage Trial of Ultra-high-Dose Methylcobalamin for ALS (JETALS), a new drug application for a high-dose formulation of mecobalamin (development code: E0302) for ALS, has been initiated [133]. Although the mechanism of action of mecobalamin in the pathogenesis of ALS has not been elucidated, results from non-clinical studies suggest its efficacy through neuroprotective and neuroaxonal regeneration effects [133].

### 5.2. Addressing DRG Neuron Abnormalities May Treat Motor Neuron Impairments in ALS

Several ALS preclinical studies have reported abnormalities in DRG neurons that precede motor neuron abnormalities [95,116]. Held et al. reported that non-motor neuron dysfunction in animals with early-stage ALS correlates with motor deficits in the absence of motor neuron degeneration, and that the ALS phenotype could be alleviated via the activation of neurotrophic BMP pathways in non-motor neurons using a knock-in SOD1-ALS model [84]. DRG neurons are beginning to attract attention as a therapeutic target for ALS movement disorders in terms of whether intrinsic receptive fibres may play a role in neurodegeneration via glutamatergic excitatory inputs to motor neurons.

The most basic neural circuit of the spinal reflex involving DRG neurons is the stretch reflex arch, which consists of Ia fibres and alpha motor neurons. Ia fibres entering from the dorsal root of the spinal cord make excitatory synaptic connections to alpha motor neurons in the anterior horn of the spinal cord. A lack of afferent input from Ia fibres to spinal gamma motoneurons has been suggested as a mechanism for disease resistance in mouse models of ALS, with some reports suggesting that reduced Ia endoreceptor muscle spindle afferents partially contribute to alpha motoneuron survival [35,134,135], and Guo et al. noted that the cytoplasmic vacuolation of DRG neurons may contribute to motor neuron degeneration in hSOD1G*93*A mice [90]. Yan et al. noted that calcium modulation led to a decrease in Ca current via HVA Ca channel-targeting abnormalities in a study in which serum from healthy adults, patients with other neurological diseases, or patients with sporadic ALS was injected into the DRG of mice for 3 days [91]. In DRG neurons, the regulation of calcium homeostasis may be achieved through drug development and interventions that restore or stabilize intracellular calcium levels, which may not only alleviate the sensory deficits associated with ALS, but may also improve motor deficits. In addition, Vaughan et al. reported that the accumulation of misfolded TDP-43 and SOD1 directly affects DRG neurons, resulting in sensitisation to stress; since the degeneration of DRG terminals approximates that seen in motor neurons, future therapies should focus on targeting these proteins [84].

These findings suggest that DRG neurons may be a potential therapeutic target, not only for ameliorating sensory abnormalities in patients with ALS, but also for treating motor impairments in ALS. Whether the normalisation of DRGs in ALS normalises motor neurons, prevents muscle atrophy, and improves motor function is currently unknown, but addressing DRG abnormalities should be considered as a novel strategy for treating ALS.

### 5.3. Potential for Gene Therapy Targeting TDP-43

Genetic abnormalities are found in DRG as well as in motor neurons in ALS (Table 4). To date, SOD1G*93*A has received the most attention as an ALS-inducing factor and has been reported in motor neurons in various studies. Preclinical studies have also reported the effects of SOD1G*93*A in DRGs [8,33,34], and though it is still considered a therapeutic target, TDP-43 accumulation has also recently attracted attention. For example, Vaughan et al. reported that the accumulation of pathological features caused by TDP-43 A*315*T expression may contribute directly to impaired neurite outgrowth in DRG [84].

TDP-43 performs several mRNA-related processes in the nucleus, including transcription, splicing, the maintenance of RNA stability, and the processing of miRNAs and long non-coding RNAs [136]. TDP-43 proteinopathy refers to diseases involving TDP-43, including ALS, fronto-temporal lobar degeneration, primary lateral sclerosis, and progressive amyotrophic disease [136]. It has been suggested that TDP-43 plays a pivotal role in the pathogenesis of ALS. TDP-43 is primarily localised to the nucleus, but also relocates to the cytoplasm for some of its functions [137]. In ALS, an increase in cytoplasmic TDP-43 levels is observed, which results in the formation of cytoplasmic inclusion bodies [137,138]. Rumpf et al. reported that autosomal dominant mutations in human valosin-containing protein cause FTD and ALS, a hallmark of which is the formation of TDP-43 aggregates [89]. Park et al. reported that photogenetic stimulation enhanced the nuclear translocation of red fluorescent protein TDP-43 accumulated in the cytoplasm in C4da neurons, a class IV dendritic branch of Drosophila larvae, through the action of calcium-dependent regulators and nuclear translocation components [88]. Nagano et al. focused on mRNA transport to neuroaxons, one of the functions of TDP-43 in ALS motoneurons, and identified multiple ribosomal protein mRNAs as target mRNAs transported to axons by TDP-43 [121]. Furthermore, ribosomal proteins are reduced in lesional axonal regions in patients with ALS, and ALS genetic therapy is underway to express ribosomal proteins using adeno-associated virus (AAV) vectors [139]. If a similar decrease occurs in the DRG due to TDP-43 accumulation, gene therapy using AAV vectors expressing ribosomal proteins may be feasible for the DRG in ALS (Figure 2).

### 5.4. Treatment Strategies for ALS Masticatory Abnormalities through the MesV

MesV neurons are proprioceptive sensory neurons that are required for mastication that not only transmit intrinsic sensations from muscle spindles in jaw movements to the centre, but also directly act on trigeminal motor neurons to control masticatory movements [35,96,97,98]. Treatment strategies for mastication disorders in ALS may focus on protecting the survival and function of the MesV, which controls mastication movements [35,36,140,141,142]. Seki et al. reported excessively inhibited and irregular neuron-firing activity and a significant decrease in the Na current in the MesV in mouse neonates [115]. Notably, MesV abnormalities are more pronounced and definitive than the electrophysiological abnormalities in trigeminal motoneurons in neonatal ALS mice [84]. As SOD1G93A mouse neonate models were used in both studies, it is possible that the abnormalities in the MesV preceded the abnormalities in the trigeminal motoneurons.

Reductions in the Na current reported in ALS model mice are associated with the suppression of MesV activity [35,142]. Moreover, the modelling of Na current inputs to the MesV using dynamic clamps has been reported to improve irregular firing activity [35,142]. Thus, the modification of Na channels can normalise MesV activity. Using fluorescent immunohistochemistry, the reduced Nav1.6 Na current observed in ALS mice was found to be due to the down-regulation of Nav1.6 Na channels on MesV neuronal membranes [35]. Specifically, the development of Na channel modifiers and the identification of factors involved in Na channel dysfunction can be considered. Notably, the abnormality reported in trigeminal motor neurons is overexcitation, the exact opposite of the overinhibition reported in the MesV. Since overexcitation is found in many motor neurons and riluzole, an ALS drug, is an inhibitor of neuronal excitability, the treatment of ALS targeting the MesV should show the opposite effect of the current treatment [128]. Another possible ALS treatment is neuropeptide Y (NPY); Seki et al. reported that the application of NPY shortens the post-hyperpolarisation duration after action potentials in the rat MesV and increases the mean spike frequency during repetitive discharges [139]. In addition, the burst frequency increases in neurons that exhibit rhythmic burst discharges in response to maintained current injections, and NPY treatment increases the Na current in the MesV [139]. Future studies should investigate whether neuropeptides that act in an excitatory capacity on neurons, such as NPY and Orexin, ameliorate the hypo-excitation of the MesV in ALS and improve MesV function [96]. Whether correcting abnormalities in proprioceptive sensory neurons involved in muscle spindles, such as the MesV, corrects mastication deficits in ALS and helps to treat movement disorders in patients with ALS should also be examined [95] (Figure 3).

## 6. Conclusions

In clinical and preclinical studies, emerging evidence suggests that both sensory and motor neurons are involved in the pathogenesis of ALS. Therefore, continued research on both neurons is needed to improve our understanding of the diagnosis, treatment, and pathogenesis of ALS. Primary sensory neurons not only transmit sensory information from muscle spindles, but are also involved in the control of movement, highlighting them as potential therapeutic targets for movement disorders in ALS. Furthermore, as shown in this review, primary sensory neurons such as DRG and MesV neurons may show abnormalities prior to motor neurons in clinical studies. As such, future therapies targeting primary sensory neurons may improve the quality of life of patients with ALS.

## Figures and Tables

**Figure 1 biomedicines-11-02967-f001:**
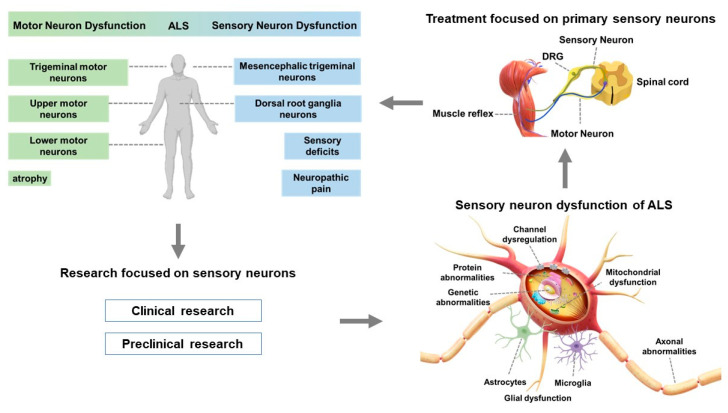
This review focuses on sensory neuron abnormalities in amyotrophic lateral sclerosis (ALS). As various sensory neuron abnormalities have been reported in studies using animal models, this review proposes a novel therapeutic strategy for ALS by correcting the proprioceptive sensory neuron abnormalities in patients with ALS.

**Figure 2 biomedicines-11-02967-f002:**
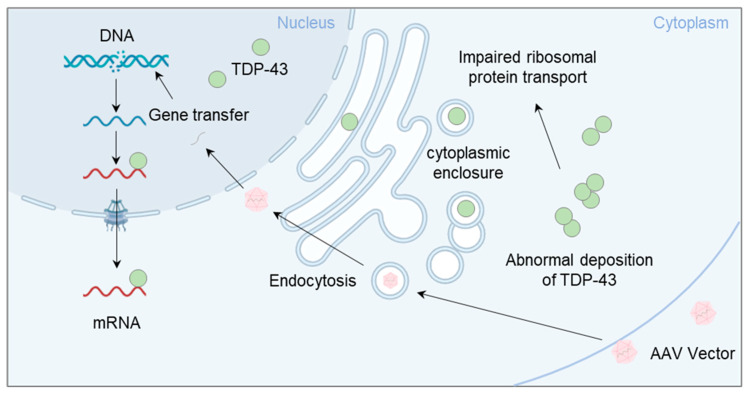
Therapeutic strategies for the cytoplasmic deposition of TDP-43 using AAV vectors. Cytoplasmic deposition of TDP-43 in ALS has been suggested to be due to abnormal RNA metabolism. Attempts to ameliorate TDP-43 cytoplasmic deposition via gene therapy using AAV vectors have been investigated and may be applicable to the treatment of DRGs.

**Figure 3 biomedicines-11-02967-f003:**
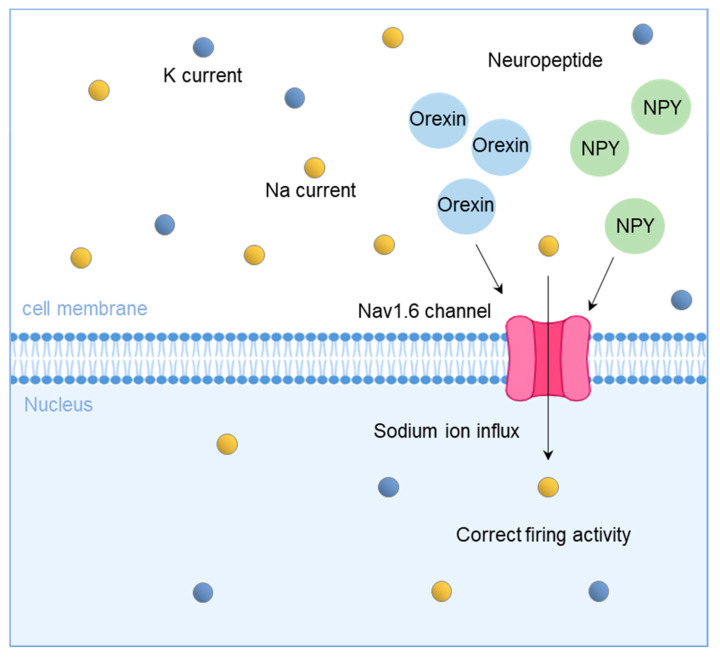
Therapeutic strategies for the correction of sodium ion channels via neuropeptides. Neuropeptides such as orexin and NPY may increase MesV sodium current and improve firing activity in ALS with reduced sodium currents.

**Table 1 biomedicines-11-02967-t001:** Major clinical studies supporting sensory involvement in ALS.

Type of Sensory Neuron	Dysfunction	Number of Patients with ALS	References
Median nerve	Attenuation of postsynaptic high-frequency somatosensory evoked potential bursts in patients with ALS with a long duration of the disease.	20	[73]
Sensory system	A total of 14.7% of patients with ALS have damaged sensory systems.	150	[62]
Sensory cortex	Sensory cortex overexcitation is associated with shorter survival in patients with ALS.	145	[71]
Intraepidermal nerve fibres	Decreased intraepidermal nerve fibres density with aggregation of TDP-43 in patients with ALS.	18	[74]
Epidermal nerve fibres	Loss of intraepidermal nerve fibres.	41	[60]
Intraepidermal nerve fibres	Increased axonal expansion rate and negative growth-related proteins in intraepidermal nerve fibres.	32	[58]
Intraepidermal nerve fibres	In 85% of patients with ALS, quantitative sensory testing showed abnormal thermal pain thresholds and skin biopsies showed decreased intraepidermal nerve fibre density.	24	[75]
Ascending sensory fibres	Anatomical damage to ascending sensory fibres in 60% of patients with solitary ALS.	21	[76]
Leg sensory nerve	A total of 27% of patients with ALS had abnormal action potential. Amplitude, and 91% had pathological abnormalities of the leg sensory nerve.	103	[56]
Sensory neuron in sural	Sensory neuropathy and axonal degeneration.	5	[77]
Median, radial, and sural nerves	Asymptomatic decline in sensory nerve function.	19	[67]
Sensory nerves	Early axonal atrophy, increased remyelination, and predominance of smaller fibre diameters.		[78]

ALS, amyotrophic lateral sclerosis; TDP-43, TAR DNA-binding protein 43.

**Table 2 biomedicines-11-02967-t002:** Characteristics of animal models of ALS with reported abnormalities of sensory neurons.

Animal Model	Animals	Characteristics of the Model	References
SOD1G*93*A	Mouse	Marked muscle weakness is observed at 15 weeks of age in SOD1G*93*A transgenic mice with familial ALS	[33]
dSOD1G*85*R	*Drosophila*	SOD1G*85*R knock-in model shows severe motor deficits with apparent degeneration of motor neurons, providing a better understanding of the contribution of multiple cell types in ALS	[72]
C*9*orf*72* ALS	*Drosophila*	Transgenic *Drosophila* model overexpressing human TDP-43 shows reduced lifespan, reduced motor activity, increased morphological defects in motor neurons, a loss of neurons, and axonal damage	[88]
TDP-43A*315*T	Mouse	A*315*T mutant TDP-43 transgenic mouse model with marked early-onset progressive neurodegeneration resulting in reduced motor performance, spatial memory and deinhibition, and reduced grip strength due to muscle atrophy	[84]
TDP-43 (TBPH)	*Drosophila*	Transgenic fly that exhibits an adult locomotor defect and shares many features with human proteins	[89]

C*9*orf*72*, chromosome 9 open reading frame 72; TDP-43 (TBPH), TAR DNA-binding protein 43; SOD1, superoxide dismutase 1.

**Table 3 biomedicines-11-02967-t003:** Reports on sensory neuronal abnormalities in animal models of ALS.

Type of Sensory Neuron	Dysfunction	Type of Animals	Genes	References
MesV	Firing abnormality	Mouse	SOD1 G*93*A	[36]
DRG	Decrease in SGNF density	Mouse	SOD1 G*93*A	[33]
C4da	TDP-43 accumulation in the cytoplasm via calcium-calpain-A-importin α3 pathway	Drosophila	C*9*orf*72* ALS	[88]
DRG	SOD1 accumulation in SGCs	Mouse	SOD1 G*93*A	[8]
MesV	Firing abnormality with sodium channel dysfunction	Mouse	SOD1 G*93*A	[35]
Non-motor neuron	Neurotrophic BMP pathway	Drosophila	dSOD1G*85*R	[72]
DRG	Shorter and less complex neurites	Mouse	TDP43 A*315*TSOD1 G*93*A	[84]
Long-projection sensory neurons	Defects in MAM signalling	Mouse	TDP43 A*315*T	[93]
Olfactory bulb and retina	Neuronal vacuolisation in olfaction and vision pathways	Mouse	SOD1 G*93*A	[87]
DRG	Accumulation of a peripherin splice variant	Mouse	SOD1 G*93*A	[34]
DRG	Accumulation of misfolded protein	Mouse	dSOD1G*85*R	[94]
Ia and II proprioceptive nerve	Early disturbances in muscle spindles before motor neuron symptoms	Mouse	TDP43 A*315*T SOD1 G*93*A	[95]
C4da	Localised accumulation, predominantly of TDP-43	Drosophila	TDP-43 (TBPH)	[89]
DRG	Mitochondrial abnormalities	Mouse	SOD1 G*93*A	[90]
DRG	Reduction of high-voltage-activated Ca^2+^ current	Mouse	ICR mice injected with sera from patients with ALS	[91]

BMP, bone morphogenetic protein; C4da, class IV dendritic arborisation; DRG: dorsal root ganglia; MAM, mitochondria-associated membrane; MesV, mesencephalic trigeminal nucleus; SGC, satellite glial cell; SGNF, sweat gland nerve fibre; VNC, ventral nerve cord.

**Table 4 biomedicines-11-02967-t004:** Genetic abnormalities observed in sensory neurons in ALS.

Genes	Type of Sensory Neuron	Function	References
SOD1	DRG	Encodes an itinerant enzyme that catalyses the conversion of superoxide to hydrogen peroxide and dioxygen.	[117]
KIF1A	DRG	Encodes the kinesin 3 motor that transports presynaptic vesicle precursors and dense core vesicles.	[123]
TARDBP	DRG	Mainly localised in the nucleus and regulates RNA processing.	[122]
FUS	VNC	RNA-binding protein.	[104]

FUS, fused in sarcoma; KIF1A, kinesin family member 1A; TARDBP, TAR DNA-binding protein (TDP-43) gene.

## Data Availability

The datasets generated and/or analysed in this study have not been published but are available from the corresponding authors upon reasonable request.

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
