# Peer review of "Characteristics of Sensory Neuron Dysfunction in Amyotrophic Lateral Sclerosis (ALS): Potential for ALS Therapy"

_biomedicines, 2023, doi:10.3390/biomedicines11112967_

Round 1
Reviewer 1 Report
Comments and Suggestions for Authors
we read with interest the article by Seki about the potential role of Sensory neurons in ALS pathology which gives new insight into this disease.
the article flows nicely; however, there are comments on certain sections that need to be revisited. It seems that this article is written by different authors and then combined together which requires some checking on the different sections.
Comments:
1- Sections pertaining to mitochondrial dysfunction is truncated and more is needed to take on mitochondrial dysfunction in ALS and it needs to be connected to sensory neurons as the article discusses sensory neurons and highlights this aspect.
2-the same applies when discussing Glial changes; there is lac of discussing the cross-talk of glia with neuronal entities which make this section so immature
3- same applies to the sections on oxidative stress and Excitotoxicity which are written in a very superficial way and need to be explored more.
4- a section on proteinopathy is needed as ALS involves several protein changes and requires this section and needs to be tied to sensory neurons
5- the review require more table of the animal models of ALS -rodent ones vs. what pathology they recapitulate from humans
6- Finally when the authors discuss possible targets for Neurotherapy, can they elaborate on the what current drugs are used and what do they target example: Edaravone
Minor:
1- please check the English writing as many times ALS is defined at different sections
2-check abbreviations as TDP has been written as TDP-43 others TDP43
3- wrong sentences: Parkinson's disease, is not a non-motor symptoms
"Several non-motor symptoms, including autonomic dysregulation, Parkinson's disease, cognitive"
Sessone et al. suggested that mis-splicing of
Comments on the Quality of English Languageproof reading is needed
- please check the English writing as many times ALS is defined at different sections
2-check abbreviations as TDP has been written as TDP-43 others TDP43
3- wrong sentences: Parkinson's disease, is not a non-motor symptoms
"Several non-motor symptoms, including autonomic dysregulation, Parkinson's disease, cognitive"
Sessone et al. suggested that mis-splicing of
Reviewer 2 Report
Comments and Suggestions for Authors
In this manuscript, the authors provided a review of sensory neuron dysfunction in ALS. The review discussed the clinical changes associated with sensory neuron dysfunction in ALS, including the types of sensory symptoms experienced by patients. It also highlighted potential therapies targeting sensory neurons in ALS, such as gene therapy, stem cell therapy, and pharmacological interventions. Overall, this review provides some new insights into the characteristics of sensory neuron dysfunction in ALS, and the potential for developing therapies that target this aspect of the disease. Although this concept is very new, there are still some things that the authors still need to clarify or improve.
1. abstract: The content is a bit too brief and does not provide enough meaningful information. I recommend adding some more specific content, especially some of the potential therapeutic targets for sensory neuron dysfunction mentioned in this manuscript. Appropriately enriching the content of the abstract will be more helpful to readers.
2. Tab. 1: The authors described this as "Sessone et al. suggested that mis-splicing of peripherin may cause axonal damage to DRG neurons and motor neurons in ALS model mice and patients with ALS". However, I cannot understand at all what this tab has to do with "therapeutic strategies". In fact, I think Tab. 1 is more suitable to be placed in the third part, because most of the content is explaining sensory neuronal abnormalities.
3. Most of the evidence comes from animal experiments and in vitro models. However, the evidence mentioned above is not very powerful in EBM. If possible, some relevant content of clinical research should be added as evidence.
4. In the fourth part, although the authors proposed several potential therapeutic strategies, their clinical value and uniqueness in improving sensory neuron dysfunction are not clear. For example, in gene therapy, how to target sensory neuron treatment, and what specific help it may have for ALS patients after successful treatment, especially the content provided is actually difficult to understand. In addition, I suggest that this section should also provide a scheme figure or chart to reinforce the relevant content.
5. Although sensory neuron abnormalities are indeed symptoms of ALS, in my experience not everyone will have obvious symptoms (no more than 1/3). In addition, symptoms the of even sensory neuron are usually not serious, at least compared to fatal symptoms of ALS. Therefore, I think the authors must explain this and explain the need for treatment of sensory neuron dysfunction. At the same time, it would be better if some clinical opinions or evidence could be provided (the evidence from animal experiments is still insufficient).
Comments on the Quality of English LanguageEnglish writing is fine.
Reviewer 3 Report
Comments and Suggestions for Authors
This is a review that provides a very concise description of ALS-related sensory neuron dysfunction, focusing on its basic mechanisms mainly on ALS mouse models, with a very superficial description of potential ALS therapies targeting sensory neurons. Although the aim of the work was carried out reliably and the data contained therein were supported by well-selected references, the article does not present any new revelations or concepts that would distinguish it from those cited in the literature list.
Major flaws:
The Abstract does not include the particular conclusions except the statement that sensory neuron abnormalities in ALS have hitherto received little attention, focusing on them will help elucidate the disease mechanism and develop effective treatments.
Introduction:
line 45 – Please change „brain” to „braistem”
…”Although the exact cause of ALS is not fully understood, it is believed to be a combination of genetic and environmental factors [39,40].”… - Please elaborate, what are they?
…”Neurophysiological studies have also demonstrated abnormalities of in sensory nerve conduction parameters, including prolonged sensory nerve action potentials and reduced sensory nerve action potential amplitudes [64–69]”…. – The role of the neurophysiological diagnostic is important in differential diagnostics of ALS from other mono- or polyneuropathies. Please expand this problem.
Please develop the meaning of the clinical examinations (what are they) currently preferred in the evaluation of the sensory dysfunction in ALS. Studies on animals are described perfectly but the clinical aspects are described only partially.
The same above refers to the treatment strategies for ALS in patients, the article is, in general, dedicated to the results of the experimental studies. The title of this article suggests something different.
The article is one great Introduction section.
There is not in fact Discussion section with the suggestions to the treatment advances or hypotheses.
The conclusions are very general and do not present any revelations or important content presented in the article.
I will suggest reducing 1/3 of the basic studies refs. towards clinically related.
Comments on the Quality of English LanguageMinor revisions required
Round 2
Reviewer 1 Report
Comments and Suggestions for Authors
Accept in present form
Reviewer 2 Report
Comments and Suggestions for Authors
In response to the questions I raised, the authors have provided appropriate explanations and corrections.
Reviewer 3 Report
Comments and Suggestions for Authors
The authors have responded to all of my queries and suggestions well. The clinical scope regarding ALS has been greatly improved, and many new references related to the topic undertaken in the paper have appeared, which increased its scientific value.